# Investigation into the Performance of NiTi Shape Memory Alloy Wire Reinforced Sn-Bi Self-Healing Metal Matrix Composite

**DOI:** 10.3390/ma15092970

**Published:** 2022-04-19

**Authors:** Nathan Salowitz, Shobhit Misra, Muhammad Istiaque Haider, Marco Povolo, Pradeep Rohatgi

**Affiliations:** 1Department of Mechanical Engineering, University of Wisconsin—Milwaukee, 3200 N Cramer Street, Milwaukee, WI 53211, USA; haiderm@uwm.edu; 2Slunca Engineers Private Limited, New Delhi 110091, India; shobhitkmishra@gmail.com; 3Department of Industrial Engineering, University of Bologna, 40132 Bologna, Italy; marco.povolo2@unibo.it; 4Department of Materials Science & Engineering, University of Wisconsin—Milwaukee, 3200 N Cramer Street, Milwaukee, WI 53211, USA; prohatgi@uwm.edu

**Keywords:** metal–matrix composites (MMCs), off-eutectic solder alloys, fiber–matrix interface, shape restoration, self-healing, shape memory alloys (SMAs)

## Abstract

Self-healing materials have the potential to create a paradigm shift in the life cycle design of engineered structures, by changing the relation between material damage and structural failure, affecting structures’ lifetime, safety, and reliability. However, the knowledge of self-healing capabilities in metallic materials is still in its infancy compared to other material systems because of challenges in the synthesis of organized and complex structures. This paper presents a study of a metal matrix composite system that was synthesized with an off-eutectic Tin (Sn)-Bismuth (Bi) alloy matrix, reinforced with Nickel–Titanium (NiTi) shape memory alloy (SMA) wires. The ability to close cracks, recover bulk geometry, and regenerate strength upon the application of heat was investigated. NiTi wires were etched and coated in flux before being incorporated into the matrix to prevent disbonding with the matrix. Samples were subjected to large deformations in a three-point bending setup. Subsequent thermo-mechanical testing of the composites confirmed the materials’ ability to restore their geometry and recover strength, without using any consumable components. Self-healing was accomplished through a combination of activation of the shape memory effect in the NiTi to recover the samples’ original macroscopic geometry, closing cracks, and melting of the eutectic material in the matrix alloy, which resealed the cracks. Subsequent testing indicated a 92% strength recovery.

## 1. Introduction

This paper presents the synthesis and testing of a novel self-healing metal matrix composite, with an off-eutectic Tin (Sn)-Bismuth (Bi) alloy matrix, reinforced with near equiatomic Nickel Titanium (NiTi) shape memory alloy (SMA) wires. The SMA wires were treated to prevent disbonding with the Sn-Bi matrix, confirmed with microstructural analysis. Samples were created and subjected to a large three-point bending deformation, causing matrix fracture and bulk deformation, then healing was thermally actuated. The results were evaluated with a combination of microstructural analysis and mechanical testing.

### 1.1. Motivation

Self-healing materials, which have an inherent ability to repair damage, have the potential to cause a paradigm shift in the life cycle of engineered structures, affecting design, use, maintenance, and even what constitutes the ‘end of life’. Many self-healing materials have been studied for a range of applications [1], including in automotive, civil, aerospace, and marine structures [2,3,4].

### 1.2. Background

The second law of thermodynamics states that the entropy of a closed system can only increase, which has been used to explain the deterioration of materials over time, from the growth of microcracking through to eventual fracture and failure [5]. The ability to synthesize organized meta-materials storing potential energy or healing mechanisms that can be released and activated, in chemical, thermal, and mechanical forms, has provided an opportunity to create materials that can repair damage [6]. The concept of self-healing materials has largely been inspired by biological systems that can detect and heal damage to maintain functionality [7,8,9]. Self-healing capabilities have the potential to reduce system down-time for inspection and repair, and costs of failure. Major research efforts in self-healing, to date, have been focused on polymeric and cementitious-based composites, because it is relatively easy to incorporate organized internal structures within these materials [10,11,12].

#### 1.2.1. Adhesive Based Self-Healing

The most common technique to create self-healing materials, prevalent in polymeric and ceramic-based self-healing materials, is to disperse capsules of liquid adhesive into the material [13,14]. When a crack forms and breaks open a capsule, adhesive will wick into the crack, react with a catalyst or air, and glue the crack closed. This approach is functional and relatively easy to implement; however, it has a number of weaknesses: (1) The encapsulated liquid cannot carry structural loads (tension or shear) but add weight to the bulk material structure. This diminishes the initial strength to weight ratio of the pristine material, making it less attractive for engineering applications. This is exemplified by the fact that tests of these materials regularly report healing efficiency greater than 100%, indicating that the strength of the healed structure is greater than the pristine structure. (2) The liquid’s inability to carry load introduces stress concentrations and potential crack initiation points. While it is conceptually good to have cracking start near the healing adhesive agent, this is a weakness in the material. (3) The glue simply flows into cracks and then cures, possibly solidifying with the structure in a deformed state. This itself can be a form of structural failure if the cracking deforms too far [15]. (4) Finally, the adhesive is consumed in the healing process, meaning that only a limited amount of damage over time can be healed.

#### 1.2.2. Metallic Self-Healing

Metallic self-healing materials are much less studied and harder to create, but offer potential advantages over polymeric or ceramic adhesive self-healing materials [13,16,17]. Metallic self-healing materials can: (1) Be entirely solid, meaning the entire structure is load bearing and (2) there are no intentionally introduced stress concentrations (3) multi-step self-healing incorporating shape memory alloys to align the structure and mechanisms to bond the structure back together, inspired by the way a human bone heals, and can prevent the accumulation of deformation, over multiple or large healing cycles and, thus, prevent deformation-based structural failure. (4) Self-healing in metals can be performed without consumables, meaning the process can ideally be repeated indefinitely. Further, metals generally have a higher volumetric material strength and are favored for many structural applications [13,17].

To date, there are three main directions that have been taken towards the development of self-healing in metallic materials [9,13,16,18]: (i) precipitation in alloys to close voids and cracks, (ii) embedding ceramic micro-balloons encapsulating low-melting-temperature alloys as healing agents into the body of an alloy with a higher melting temperature, and (iii) creating an off-eutectic metal alloy; when heated, eutectic portions of the material will soften and melt into the cracks, while the off-eutectic portions of the material will remain solid, maintaining structural geometry.

A bulk metal designed to heal in one of these ways still cannot recover macroscopic deformations and would potentially solidify in a deformed state that could be considered a failure. To overcome this, research has been performed to incorporate shape memory alloy (SMA) wire reinforcements into the metals. When heated above the austenite finish transition temperature, these SMA wires will try to return to their original, parent geometry, through a reversible phase transformation [19]. The integration of SMA wires (fibers) into an off-eutectic metal matrix alloy can create a self-healing material, with multiple healing steps that mimic the healing of a human bone, including alignment of the structure (setting) and then soldering of the matrix.

### 1.3. Challenges

Exploration of this combination of healing methods has been limited and inhibited by the fact that the most common SMA, Nickel Titanium (NiTi), forms an inert surface layer of titanium dioxide (TiO_2_) that is very hard to bond to or remove. In order to overcome this poor bonding property of TiO_2_, work to date has wrapped the NiTi wires around physical/mechanical anchors, such as bolts, within the material structure. This approach creates a poor composite structure because of the poor stress transfer between the fibers and matrix.

SMAs, including NiTi, also have limited heat tolerance. Exposure to too high a temperature can re-train the parent geometry or degrade the shape memory capabilities of the material. Therefore, the melting point of the matrix must not be excessive.

### 1.4. Objective

This paper presents the synthesis and testing of a NiTi wire reinforced off-eutectic Sn-Bi metal matrix composite, where the wires were etched to promote bonding between the NiTi and Sn-Bi. Sn-Bi was selected as the matrix because it is a low-temperature-melting solder. Specimens were synthesized, inspected, and their self-healing capabilities were tested.

## 2. Materials and Methods

Experimental metal matrix composite specimens were created in a multi-step process where individual NiTi SMA wires were treated to improve bonding with the Sn-Bi matrix. The treated wires were then cast into the matrix using a pressure infiltration system. A Scanning Electron Microscope (SEM) and Energy Dispersive X-Ray Spectroscopy (EDS) were used to inspect the interface between wires and matrix. The samples were mechanically deformed in a three-point bending test, to create macroscopic cracking and significant deformation of the overall specimen. After further inspection, the samples were heated on a hot plate to actuate self-healing. The healed samples were then mechanically tested again to evaluate the recovered strength.

### 2.1. Sample Synthesis

Sn-Bi matrix alloy was made from tin and bismuth ingots donated by Kohler Company (Kohler, WI, USA) which were melted together at a temperature well above the melting point of both metals (Bi, T_m_ = 271.43 °C). The molten alloy was stirred for several minutes with a graphite impeller to insure uniform distribution. Gravity castings were made to store the alloy for later use. An alloy containing Sn with 20 wt% Bi was selected to meet Manuel’s recommendation that 15–20 wt% liquid would be optimum to allow free flow of inter-dendritic fluid, essential to the healing process [4]. The Bi-Sn phase diagram indicates the eutectic point at 40% Bi with a eutectic melting temperature of 138 °C. Based on the tin-bismuth phase diagram and Lever Rules the healing temperature was calculated to be 165 °C to produce the desired partial melting [20,21].

Dynalloy (Irvine, CA, USA) Flexinol© 90 NiTi wire 500 µm in diameter and with transition temperatures of: martensite start temperature, M_s_ = 75 °C, martensite finish temperature M_f_ = 65 °C, austenite start temperature, A_s_ = 90 °C, austenite finish temperature, A_f_ = 110 °C was cut to 65 mm lengths and prepared for use at the composite fiber reinforcement [22]. To improve shear transfer between the NiTi wires and matrix, the NiTi wires were processed following the method proposed by Dutta et al. and Coughlin et al. to remove the inert surface oxide layer that is a common detriment to bonding to NiTi [23,24,25]. The NiTi wires were etched in 4.8%HF-10.5% HNO_3_ aqueous solution for 3 to 5 min in an inert nitrogen environment followed by dipping the wires in a commercially available phosphoric acid-based flux, Indalloy Flux 2 (Indium Corporation, Clinton, NY, USA) [26]. The etched and fluxed wires were then dip coated with molten solder to further inhibit re-oxidation of the etched NiTi surface. Dip coating of the wires in the molten Sn-Bi alloy burned off the flux and created single-fiber composites. Groups of wires were placed in a pressure infusion system and the pressure infiltration technique adapted from Ruzek et al. [27] was used to fabricate composites with a 20% volume fraction of NiTi reinforcement. In this process the NiTi wires were cast straight, not wrapped around bolts or other mechanical anchors to pull against if bonding between the NiTi and Sn-Bi failed.

The geometry of the samples was specifically designed to allow the cracking and fracture of the matrix in a three-point bending test. Each sample had a 20% volume fraction (V_f_) of NiTi wires that were generally aligned along the length of the sample in a prismatic geometry with a length (l) of 65 mm, width (b) of 12.5 mm, and thickness (d) of 3 mm as depicted in Figure 1.

### 2.2. Pre-Test Analysis

Optical microscopy and SEM were used to examine the continuity of the interface between the tin bismuth alloy and the NiTi reinforcement and the structure at the interface. To perform this inspection the castings were ground and etched to improve the quality of microscopic images. A Hitachi S-4800 SEM was used to image the samples at high magnifications as shown in Figure 2. Back Scattered Electron (BSE) Imaging was used to identify Sn rich vs. Bi rich regions. The Bi-rich regions appear lighter compared to Sn-rich regions because the Bi is the denser of the two metals.

The SEM images showed a dendritic solidification structure in the matrix tin bismuth alloy, around the dark-appearing fiber of NiTi. The images also show continuity of contact between the matrix of tin bismuth alloy and the NiTi, without any visible porosity as shown in Figure 2. This suggests that the etching and fluxing of NiTi wire promoted continuous wetting between the matrix and the fiber during processing.

Figure 3 shows a line scan across the interface between the NiTi and BiSn and Figure 4 shows the elemental maps of the sample across the interface of the Sn-Bi and NiTi highlighting the distribution of elements in a multi-phase region of the matrix. There appears to be an extremely thin layer at the interface where nickel, titanium, tin and bismuth are present.

### 2.3. Mechanical Testing

Mechanical testing of the specimens consisted of three steps: (1) Three-point bending of the specimens, (2) thermally actuating the specimen’s healing mechanisms, and (3) repeating the three-point bending test to evaluate recovered strength.

(1)Three point bending tests were performed according to ASTM D7264 using an Instron Model 8033 Load Frame. The supports for the test were 55 mm apart (L), at the center, and the bending test was continued until the center load displaced the specimen 9.5 mm (D), creating a bend angle of 142° as shown in Figure 5 and Figure 6. This produced significant deformation in the specimen combined with surface cracking easily visible to the naked eye. During the deformation, the matrix plastically deformed and then fractured in tension on the bottom of the specimen. On the top, compressive side the matrix showed minor plastic deformation. At the same time, the NiTi wires experienced recoverable strain wherein the initially twinned martensite was detwinned. Because of the extreme recoverable strains, NiTi SMAs can survive without true plastic deformation, so the NiTi survived and was functional while the matrix fractured. Additionally, some of the NiTi disbonded from the matrix due to a combination of differing strain mechanisms and crack bridging phenomena.(2)After deforming the specimens, they were placed on a hotplate to actuate their self-healing capability. The self-healing mechanism in this system consisted of two complementary phenomena: (A) actuation of the shape memory alloy and (B) melting and welding eutectic material in the matrix.
The thermal cycling during the dipping and casting steps, well above the austenite transition finish temperature of 115 °C in a fluid environment, forced the NiTi wires to assume their parent geometry in an unconstrained austenite state. Then the wires cooled, essentially unconstrained, to a self-accommodated twinned martensite state. From that state, the NiTi could be deformed with recoverable strains up to roughly 5% and without causing plastic deformation up to 10% during the bending test [28,29,30]. After the bending test, when heated through the reverse transformation to a temperature above the austenite transition finish temperature, the NiTi actuated to resume its parent geometry, as cast, restoring overall geometry, straightening the specimen and pulling the cracks closed. Plastic deformation of the matrix and geometric changes due to the fracture could inhibit the NiTi from resuming its parent geometry; however, the NiTi would continue to actuate until the temperature fell below the martensite finish temperature.Softening of the matrix at the elevated temperature allowed the forces created by the NiTi to return the overall specimen to its original shape. Additionally, regions of eutectic alloy in the matrix melted and solidified welding the matrix back together, restoring the continuity of the specimen and its strength upon cooling.

The healing process was performed and photographed at regular intervals to document the motion of the specimens. An angular rate of recovery $\dot{\left(\theta \right)}$ was calculated according to (1) where θ is the angle at the time (t) as shown in Figure 5, θ_0_ is the initial angle (180°) and t_0_ is the time that the specimen was placed on the hotplate.
(1)$$\dot{\theta }=\frac{\theta_{0}-\theta }{t-t_{0}}$$

The order of measurement vs. initial value in the numerator and denominator are reversed to produce a positive healing rate. Angles were measured using the ImageJ image processing and analysis software. While geometric recovery took less than one minute, the healing temperature was maintained for one hour to allow for welding. Observations of crack closure were performed visually and with an optical microscope.

(3)After healing, the three-point bending test was repeated to evaluate the strength recovery of the specimen. It was expected that some aspects of the damage, such as oxidation of exposed NiTi degrading re-bonding with the matrix, would cause slight reductions in the strength of the healed sample relative to the pristine sample. However, these effects were expected to be small.

## 3. Results and Discussion

Thermo-mechanical testing was performed to assess the self-healing capabilities of the specimens. The various material analyses demonstrated good wetting and alloying between the NiTi fibers and Sn-Bi matrix, indicating good adhesion.

The specimens were placed on a hot plate at 165 °C to activate healing. Testing was performed at standard temperature and pressure and the samples began the process at ambient temperature (20 °C). Heat transferring into and through the specimens took 38 s to reach the NiTi wires and heat them to the austenite transition start temperature (A_s_) of 90 °C, at which point the sample began to straighten, noted as T_0_. The geometric recovery continued as the sample’s temperature continued to increase. Motion stopped when the sample was completely straight and the NiTi wires exceeded the austenite transition finish temperature (A_f_) of 110 °C, taking an additional 36 s (74 s from the time the samples were placed on the hot plate). Pictures of the recovery process every 9 s are shown in Figure 6. Based on (1), the angular rate of recovery was approximately 1°/s.

The specimens were left on the hotplate at 165 °C for 1 h, to allow the metal matrix to weld back together. The welding of the cracks was observed optically. Specimens had two different initial conditions that resulted in differing surface appearances: (1) smooth surface, (2) a stress concentration notch, roughly 1 mm deep and 1.13 mm wide, to control the location of cracking.

Recovery of the microstructure can be seen in Figure 7, indicating that welding had regenerated a continuous material. Visual observation indicated that eutectic melting to weld the matrix back together began within 15 min of being placed on the 165 °C hot plate, as can be seen in Figure 8. The application of heat was maintained for 1 h in total, during which eutectic in the interdendritic regions melted and flowed to the crack, leaving shrinkage on the surface during resolidification, which appears as white spots in Figure 8.

After the specimens healed, they were subjected to another three-point bending test. The results of testing of the sample from the pristine and healed states were compared to each other, as seen in Figure 9. Another sample, composed of the Sn-Bi matrix, without NiTi reinforcement (that cannot recover geometry), was also mechanically tested for comparison, and data are included in Figure 9. Flexural Strain (ε_f_) reported is the maximum normal strain that occurs in a simply supported beam undergoing bending and is calculated according to (2), where D is the center displacement (max 9.5 mm), d is the thickness of the specimen (3 mm), and L is the distance between supports (55 mm).
(2)$$\varepsilon_{f}=\frac{6\mathrm{Dd}}{L^{2}}$$

Flexural stress (σ_f_) is equivalent to the maximum normal stress that occurs in a simply supported beam subjected to a central load (as in three-point bending) and is calculated according to Equation (3), where F is the applied force and b is the specimen width (12.5 mm).
(3)$$\sigma_{f}=\frac{3\mathrm{FL}}{2{\mathrm{bd}}^{2}}$$

The strength recovery (S%) for the composite was calculated according to (4).
(4)$$S\%=\frac{\sigma_{f\_\mathrm{healed}}}{\sigma_{f\_\mathrm{pristine}}}$$

## 4. Conclusions

Self-healing metal matrix composite specimens were created with Sn-20%Bi off-eutectic alloy, reinforced with 20% NiTi fibers (by volume). To promote bonding and shear transfer between the NiTi fibers and the matrix, the fibers were etched to remove the native TiO_2_ layer, then in an inert environment, they were dipped in flux and matrix material, creating pre-treated individual fibers. Inspection of the individual fibers with SEM/EDS revealed the presence of an inter-metallic layer, consisting of Sn-Ti-Ni at the fiber–matrix interface. The fibers were then grouped and cast into metal matrix composites using a pressure infusion system.

The self-healing capability of the samples was tested by subjecting them to a three-point bending test, to the point that macroscopic fracture and deformation were easily seen with the naked eye. The healing mechanism of the samples was activated by placing them on a hot plate at 165 °C. As heat propagated through the samples and their temperature increased, the shape memory effect was actuated in the NiTi, which straightened the specimens and closed the cracks. The restoration of geometry took 36 s and was completed 74 s after placing the sample on the hot plate. A continuing increase in temperature melted the eutectic material in the matrix, allowing it to flow into the cracks and weld the structure back together. The ability for the NiTi to restore the overall geometry, without the need for physical anchors to pull against, demonstrated the improvement of the bonding between the NiTi fibers and Sn-Bi matrix, achieved through the employed etching method. Using an off-eutectic alloy allowed this phenomenon to occur without melting the whole specimen. After healing, the specimens were cooled to room temperature and the three-point bending test was repeated. The final test found that the specimens had regained 92% of their original strength. Oxidation of the NiTi surface upon exposure to air in the damaged state may have resulted in poor bonding between the fiber and matrix during healing, potentially reducing the strength recovery. Further study is looking into this effect.

## Figures and Tables

**Figure 1 materials-15-02970-f001:**
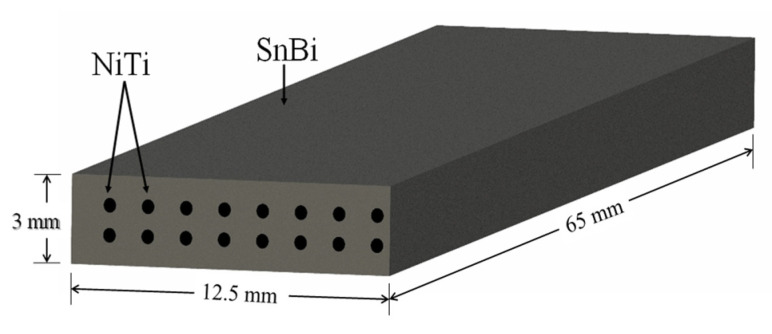
Sample Geometry.

**Figure 2 materials-15-02970-f002:**
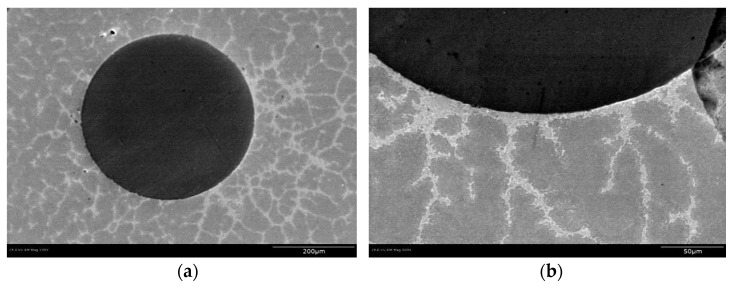
SEM image of Sn-20% Bi with a NiTi reinforcement (d = 500 μm) at (**a**) 300× and at (**b**) 1000× magnification. Reprinted/adapted with permission from Ref. [20]. 2013, Shobhit Misra.

**Figure 3 materials-15-02970-f003:**
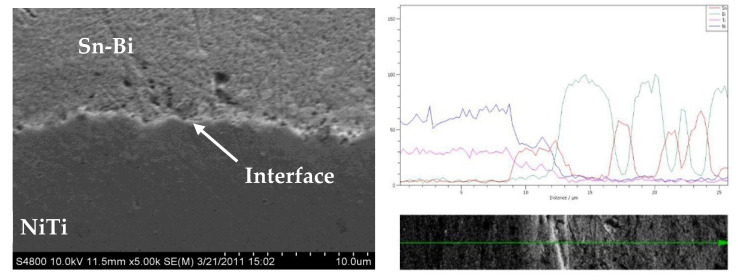
X-ray Spectrographic scan across the material interface. Reprinted/adapted with from Ref. [20]. 2013, Shobhit Misra.

**Figure 4 materials-15-02970-f004:**
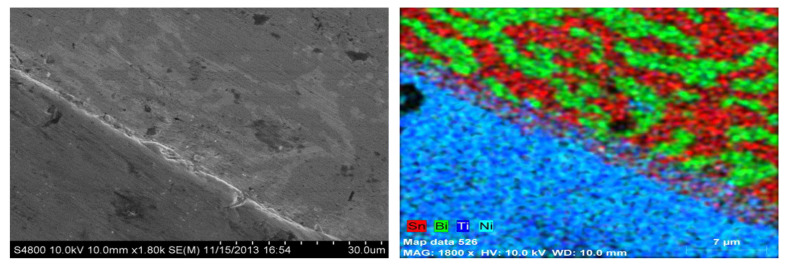
Elemental maps of the interface between Sn-Bi and NiTi. Reprinted/adapted with permission from Ref. [20]. 2013, Shobhit Misra.

**Figure 5 materials-15-02970-f005:**
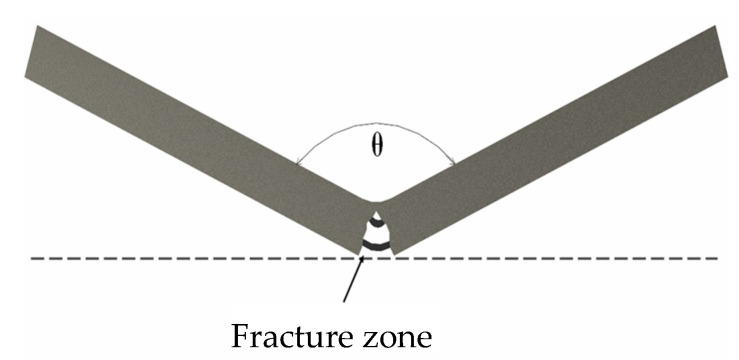
Deformed sample geometry.

**Figure 6 materials-15-02970-f006:**
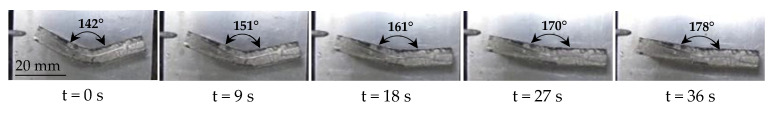
A Sn-Bi specimen with NiTi reinforcement on a 165 °C hotplate undergoing angular recovery. Pictures are at 9 s intervals from the initial motion the point where motion stopped. Reprinted/adapted with permission from Ref. [20]. 2013, Shobhit Misra.

**Figure 7 materials-15-02970-f007:**
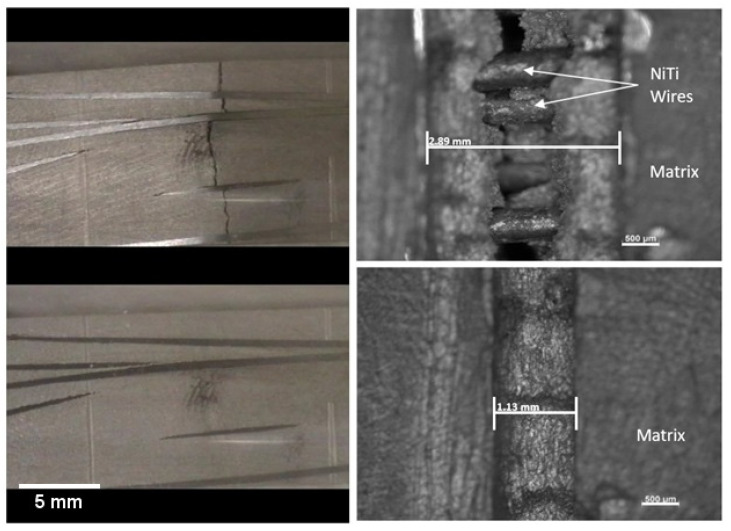
Photographs of smooth (**left**) and notched (**right**) specimens in cracked (**top**) and healed (**bottom**) states. Reprinted/adapted with permission from Ref. [20]. 2013, Shobhit Misra.

**Figure 8 materials-15-02970-f008:**
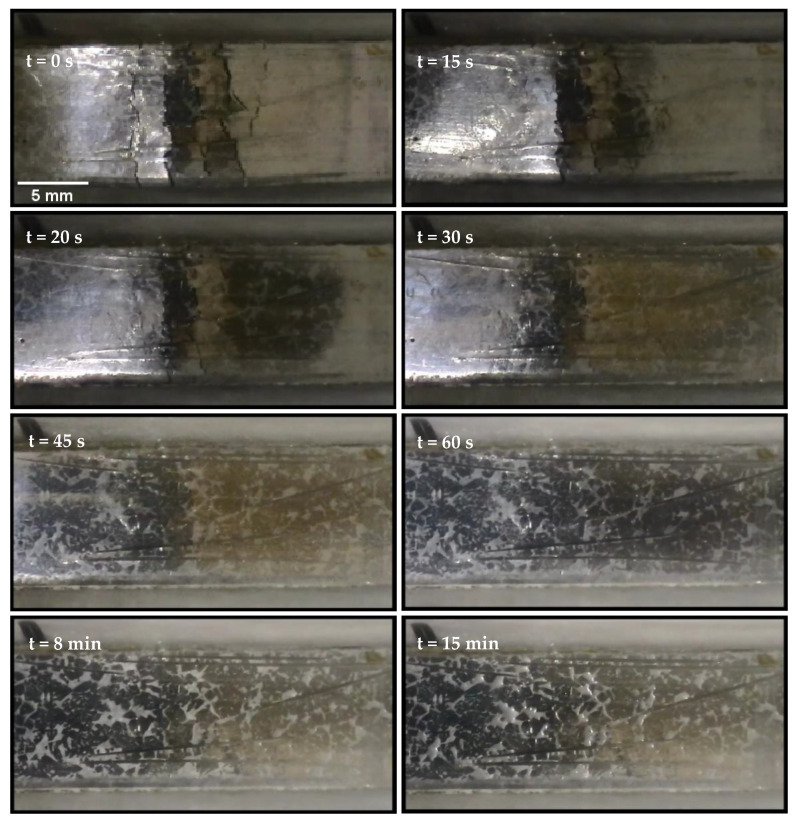
Healing of microstructure in Bi-20% Sn. In the later stages, the molten eutectic phase can be seen to sink in the last stage. This fills any internal cracks in the matrix. Reprinted/adapted with permission from Ref. [20]. 2013, Shobhit Misra.

**Figure 9 materials-15-02970-f009:**
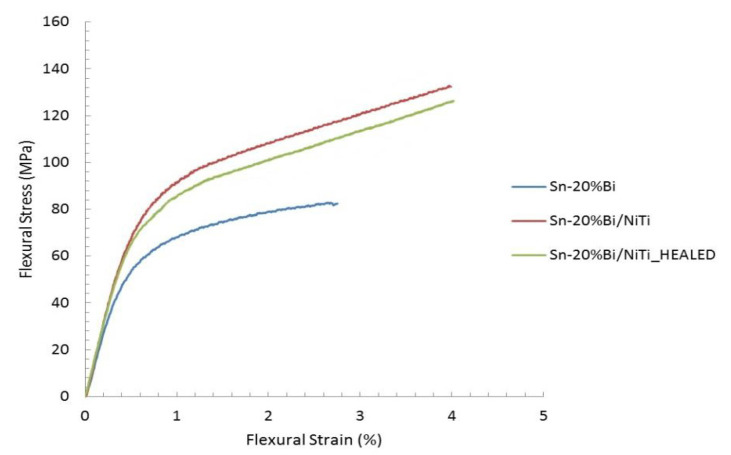
Stress strain relation measured for a pristine sample, healed sample, and a sample composed entirely of matrix material. Reprinted/adapted with permission from Ref. [20]. 2013, Shobhit Misra.

## Data Availability

The data presented in this study are available on request from the corresponding author. At the time the project was carried out, there was no obligation to make the data publicly available.

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
