# Peer review of "Investigation into the Performance of NiTi Shape Memory Alloy Wire Reinforced Sn-Bi Self-Healing Metal Matrix Composite"

_materials, 2022, doi:10.3390/ma15092970_

Round 1
Reviewer 1 Report
The manuscript reports on the investigation of self-healing behavior of NiTi-wire reinforced Sn-Bi metal matrix composite. The manuscript has an interesting subject. However, the figures and data in this manuscript were just reprinted from Ref. 26. Therefore, it is hard to consider a research article for publication.
Author Response
This is the first time this work is being published in a scientific peer reviewed journal.
Reviewer 2 Report
This paper presents a study of a metal matrix composite system that was synthesized with an off-eutectic tin - bismuth alloy matrix reinforced with nickel-titanium shape memory alloy wires. The ability to close cracks, recover bulk geometry, and regenerate strength upon the application of heat was investigated which showed that 92% of strength recovered. It is an interesting paper an reported the novel properties of self-healing effect. It should be suitable for publication on Materials. However, a few points should be elaborated more:
- In this work, the off eutectic SnBi compound is used. How far from the eutectic point or what is the degree of off-eutectic? After healing, at weld joint, or crack closing, what is degree of off-eutectic after healing?
- During the bending test, the NiTi will expose to the air, then, after healing, will the bonding between Sn-Bi and NiTi become poorer, so that the final strength is poorer after bending test. Can author make more statement on this point?
- In Figure 4, the meaning of each figure is not well described and discussed. Why so many figures needed to present the element distributions. And the scale are too small.
- Several typos and spelling can be considered to be changed:
- Line157: Volume fraction should be volume fraction?
- Line 145 &167: Tin Bismuth should be tin bismuth?
- In the Figure caption of Figure 2: 500mm should be 500 m One blank space is missed between number and unit.
Author Response
Thank you for the reviewer comments. The authors have made modifications to the manuscript in order to address them, as highlighted below/attached and the version of the manuscript with changes tracked.
- Comment: In this work, the off eutectic SnBi compound is used. How far from the eutectic point or what is the degree of off-eutectic? After healing, at weld joint, or crack closing, what is degree of off-eutectic after healing?
Response:
Information about the eutectic point of this alloy was added around line 141 as a point of reference.
- Comment: During the bending test, the NiTi will expose to the air, then, after healing, will the bonding between Sn-Bi and NiTi become poorer, so that the final strength is poorer after bending test. Can author make more statement on this point?
Response:
This was briefly discussed in point 3 of section 2.3 in the original manuscript. The section was modified and further comments relating the oxidation and potential effects on the healed strength were added in the conclusion.
- Comment: In Figure 4, the meaning of each figure is not well described and discussed. Why so many figures needed to present the element distributions. And the scale are too small.-
Response:
Figure 4 was reduced to 2 images and enlarged per recommendation. Additional discussion of the figure was added to the body of the text
- Comment: Several typos and spelling can be considered to be changed:
Response: Editing performed to correct typos
- Comment: Line157: Volume fraction should be volume fraction?
Response: Corrected
- Comment: Line 145 &167: Tin Bismuth should be tin bismuth?
Response: Corrected
- Comment: In the Figure caption of Figure 2: 500mm should be 500 m One blank space is missed between number and unit.
Response: Corrected

Reviewer 3 Report
This work used NiTi shape memory alloy to reinforce the self-healing performance of Sn-Bi metal matrix composite. Some questions need to be answered before acceptance.
- The magnification factor of Fig. 4a is different from that of Fig. 4b-4f.
- Why is the same element repeated in different pictures of Figure 4?
- Fig. 6, the angle change should be pointed out for better comparison. Besides, the image resolution is too low and the fracture zone is hard to see.
- Fig. 8, there are many white species shown after 45 s of heating. Please explain.
- Why choose 165 oC as the heating temperature?
Author Response
Thank you for the reviewer comments. The authors have made modifications to the manuscript in order to address them, as highlighted below/attached and the version of the manuscript with changes tracked.
- Comment: The magnification factor of Fig. 4a is different from that of Fig. 4b-4f.
Response: clear scale bars are visible on the 2 remaining subfigures.
- Comment: Why is the same element repeated in different pictures of Figure 4?
Response: The number of subfigures was reduced and enlarged to eliminate this.
- Comment: Fig. 6, the angle change should be pointed out for better comparison. Besides, the image resolution is too low and the fracture zone is hard to see.
Response: Higher resolution pictures have been included with angle measurements highlighted.
- Comment: Fig. 8, there are many white species shown after 45 s of heating. Please explain.
Response: Discussion of this was added around line 275..
- Comment: Why choose 165 oC as the heating temperature?
Response: Section 2.1 was modified to address this with the following:
“An alloy containing Sn with 20 wt% Bi was selected to meet Manuel’s recommendation that 15-20 wt% liquid would be optimum to allow free flow of inter-dendritic fluid, essential to the healing process [4]. The Bi-Sn phase diagram indicates the eutectic point at 40% Bi with a eutectic melting temperature of 138° C. Based on the tin-bismuth phase diagram and Lever Rules the healing temperature was calculated as to be 165°C to produce the desired partial melting.”